# Report

# OX40 ligand newly expressed on bronchiolar progenitors mediates influenza infection and further exacerbates pneumonia

Taizou Hirano[1], Toshiaki Kikuchi[1,*,†], Naoki Tode[1], Arif Santoso[1], Mitsuhiro Yamada[1], Yoshiya Mitsuhashi[1], Riyo Komatsu[1], Takeshi Kawabe[2], Takeshi Tanimoto[3], Naoto Ishii[2], Yuetsu Tanaka[4], Hidekazu Nishimura[5], Toshihiro Nukiwa[1], Akira Watanabe[6] & Masakazu Ichinose[1]

## Abstract

Influenza virus epidemics potentially cause pneumonia, which is responsible for much of the mortality due to the excessive immune responses. The role of costimulatory OX40–OX40 ligand (OX40L) interactions has been explored in the non-infectious pathology of influenza pneumonia. Here, we describe a critical contribution of OX40L to infectious pathology, with OX40L deficiency, but not OX40 deficiency, resulting in decreased susceptibility to influenza viral infection. Upon infection, bronchiolar progenitors increase in number for repairing the influenza-damaged epithelia. The OX40L expression is induced on the progenitors for the antiviral immunity during the infectious process. However, these defense-like host responses lead to more extensive infection owing to the induced OX40L with α-2,6 sialic acid modification, which augments the interaction with the viral hemagglutinin. In fact, the specific antibody against the sialylated site of OX40L exhibited therapeutic potency in mitigating the OX40L-mediated susceptibility to influenza. Our data illustrate that the influenza-induced expression of OX40L on bronchiolar progenitors has pathogenic value to develop a novel therapeutic approach against influenza.

**Keywords** bronchioles; glycosylation regeneration; OX40 ligand; viral pneumonia
**Subject Categories** Microbiology, Virology & Host Pathogen Interaction

## Introduction

Influenza viruses are RNA viruses belonging to the *Orthomyxoviridae* family and are responsible for seasonal epidemics that yearly cause 3-5 million clinical infections and 250,000–500,000 deaths (Barik, 2012; Kuiken *et al*, 2012; van de Sandt *et al*, 2012; Hayden, 2013). Recognizing that excessive immune responses are associated with the life-threatening immunopathology of influenza, several studies have been conducted to better understand the contributions of costimulatory molecules in regulating the host immune response during influenza infection (Kim *et al*, 2011; Barik, 2012; Braciale *et al*, 2012; Damjanovic *et al*, 2012). OX40 (also known as CD134, TNFRSF4) is a 50-kDa type 1 transmembrane protein, which is predominantly expressed on activated T cells and has a costimulatory function promoting T-cell proliferation and survival (Watts, 2005; Cavanagh & Hussell, 2008; Croft, 2010; Goulding *et al*, 2011). In a mouse model of sublethal influenza infection, blockade of OX40 costimulation has been shown to reduce T-cell accumulation within the lung and to diminish the destruction of lung tissue, which is correlated with the prevention of weight loss (Kopf *et al*, 1999; Humphreys *et al*, 2003).

The OX40's binding partner, OX40 ligand (OX40L, also known as gp34, CD252, TNFSF4), is a type II glycoprotein of 183 amino acids with 133 extracellular amino acids in the carboxyl terminus (Croft, 2010). The OX40L expression is constitutively low, but can be induced preferentially on professional antigen-presenting cells for T-cell priming via OX40 engagement after antigen recognition (Watts, 2005; Croft, 2010). Additional findings that non-hematopoietic cell types such as endothelial cells and smooth muscle cells also have the potential to express OX40L suggest that OX40–OX40L interactions can participate in several aspects of the physiological links between T cells and non-hematopoietic cells (Imura *et al*, 1996; Burgess *et al*, 2004; Croft *et al*, 2009).

1  Department of Respiratory Medicine, Tohoku University Graduate School of Medicine, Sendai, Japan
2  Department of Microbiology and Immunology, Tohoku University Graduate School of Medicine, Sendai, Japan
3  Kanonji Institute, The Research Foundation for Microbial Diseases of Osaka University, Kanonji, Japan
4  Department of Immunology, Graduate School of Medicine, University of the Ryukyus, Okinawa, Japan
5  Virus Research Center, Sendai Medical Center, National Hospital Organization, Sendai, Japan
6  Research Division for Development of Anti-Infective Agents, Institute of Development, Aging and Cancer, Tohoku University, Sendai, Japan
  *Corresponding author. Tel: +81 25 368 9321; Fax: +81 25 368 9326; E-mail: kikuchi@med.niigata-u.ac.jp
  †Present address: Department of Respiratory Medicine and Infectious Diseases, Niigata University Graduate School of Medical and Dental Sciences, Niigata, Japan

In the present study, because the impact of OX40 as well as OX40L on influenza viral infection has not yet been extensively investigated in lethal disease situations, we examined whether blocking OX40–OX40L interactions can actually reduce the mortality of influenza-infected mice. We found that deficiency of OX40L, but not that of OX40, markedly improved the survival in spite of the influenza A viral burden. This unexpected dissimilarity between OX40 and OX40L was thought to result from the influenza virus-binding capacity of OX40L on bronchiolar progenitors, which increased the cell numbers upon influenza infection to mediate epithelial repair and induced the OX40L expression on their surfaces to stimulate immune responses.

# Results

### Severity of lethal influenza pneumonia depends on OX40L rather than OX40

Blocking OX40 engagement has been shown to prevent immune-mediated lung damage in a sublethal influenza infection model by eliminating the influenza-induced CD4$^+$ and CD8$^+$ T-cell infiltration within the lung (Kopf *et al*, 1999; Humphreys *et al*, 2003; Croft, 2010). To dissect the implications of OX40–OX40L interactions for severe infection, we used a lethal influenza A pneumonia model for OX40L-deficient (OX40L$^{-/-}$) and OX40-deficient (OX40$^{-/-}$) mice (Fig 1A–C). Consistent with previous reports of sublethal influenza infection, OX40$^{-/-}$ mice showed decreases in the total numbers of BAL cells and amounts of cell infiltrate to the lungs as compared with wild-type mice ($P < 0.01$, Fig 1B and C). However, the survival of OX40$^{-/-}$ mice was comparable to that of wild-type mice ($P > 0.1$, Fig 1A). Comparable levels between OX40$^{-/-}$ mice and wild-type mice were also observed for interleukin (IL)-4, IL-6, and interferon (IFN)-$\gamma$ in BAL fluids (IL-4, $P > 0.3$, Appendix Fig S1A; IL-6, $P > 0.8$, Appendix Fig S1B; IFN-$\gamma$, $P > 0.5$, Appendix Fig S1C). Interestingly, OX40L$^{-/-}$ mice significantly survived the lethal influenza A/H1N1 pneumonia as compared with OX40$^{-/-}$ mice and wild-type mice, showing decreased total numbers of BAL cells, amounts of lung leukocyte infiltration, and cytokine levels in BAL fluids except for IL-4 ($P < 0.001$, Fig 1A; $P < 0.005$, Fig 1B and C; IL-4, $P > 0.8$, Appendix Fig S1A; IL-6, $P < 0.0001$, Appendix Fig S1B; IFN-$\gamma$, $P < 0.05$, Appendix Fig S1C). Similar results were achieved using influenza A/H3N2 virus and reducing the viral load of influenza A/H1N1 by half, from 5 to 2.5 times the minimal lethal dose (OX40L$^{-/-}$ vs. OX40$^{-/-}$: $P < 0.05$, Appendix Fig S2A; $P < 0.005$,

day 6, Appendix Fig S2B; $P < 0.001$, Appendix Fig S3A; $P < 0.01$, day 6, Appendix Fig S3B). These results suggest that severe influenza pneumonia can be attributed to a biological property of OX40L that is not associated with the OX40 triggering.

### OX40L on non-hematopoietic cells contributes to the severity

We next investigated which of the hematopoietic and non-hematopoietic cells serve pathogenic OX40L for the lethal influenza A pneumonia (Fig 1D–F). After bone marrow reconstitution with wild-type or OX40L$^{-/-}$ donor cells, wild-type and OX40L$^{-/-}$ recipient mice were challenged with a lethal dose of influenza A/H1N1 virus. Regardless of the donor cell type, wild-type recipient mice lost over 25% of their body weight at the start of the experiment, whereas OX40L$^{-/-}$ recipient mice did not (wild-type recipients vs. OX40L$^{-/-}$ recipients on day 7: wild-type donor, $P < 0.05$; OX40L$^{-/-}$ donor, $P < 0.05$; Fig 1D). Similar results were achieved in the differential cell counts in BAL and assessment of lung histology with the viral burden (Fig 1E and F), showing that a deficiency of OX40L on non-bone marrow-derived (i.e., non-hematopoietic) cells results in less viral production and less lung inflammation compared to that on bone marrow-derived (i.e., hematopoietic) cells.

### Increased number of OX40L-positive lung cells including bronchiolar progenitors

To characterize non-hematopoietic cells expressing OX40L, which has a key role in regulating the responses, we undertook a flow cytometry approach to analyze OX40L-positive cells in lungs exposed to a lethal load of influenza A virus (Fig 2A–C). The flow cytometry data revealed that the frequency and the number of OX40L-positive (OX40L$^{pos}$) lung cells increased in influenza-infected mice (more than sixfold, $P < 0.0001$, Fig 2A). The OX40L$^{pos}$ lung cells were fractionated using the endothelial marker CD31 and the pan-hematopoietic marker CD45, and one quarter of them were negative for both markers (Fig 2B). When the early hematopoietic/pan-endothelial marker CD34 was further included among the markers for the lineage-negative selection (Lin$^{neg}$, CD31$^{neg}$CD45$^{neg}$CD34$^{neg}$), nearly 80% of OX40L$^{pos}$Lin$^{neg}$ lung cells could be fractionated for subpopulations of bronchiolar progenitors featured as low Sca-1 and low autofluorescence (Sca-1$^{low}$AF$^{low}$, Fig 2C). These results implicate bronchiolar progenitors in the increased OX40L expression on non-hematopoietic lung cells exposed to influenza A virus.

---

**Figure 1.   Mice lacking OX40L, especially on non-bone marrow-derived cells, are more resistant to lethal influenza infection than those lacking OX40.**

A–C   Wild-type (WT), OX40L$^{-/-}$, and OX40$^{-/-}$ mice were intratracheally infected with a lethal dose of influenza A/H1N1 virus (PR8 strain). Controls included wild-type mice treated with saline (mock).

D–F   The study was similar to that in panels (A–C), but wild-type and OX40L$^{-/-}$ mice that were transplanted with wild-type or OX40L$^{-/-}$ bone marrow (BM) were used.

Data information: The susceptibility was determined by the survival of the mice ($n = 12$, A), body weight change ($n = 4$, D), total and differential cell counts in bronchoalveolar lavage (BAL; $n = 3$, B; $n = 4$, E), and histopathology of lung sections stained with hematoxylin and eosin (H&E, C and F; scale bar, 200 μm). For panels (B, D and E), data are shown as the mean ± standard error. In panel (F), lung sections were also immunostained with antibody to matrix protein 2 of influenza virus (M2, green), and the nuclei were identified with DAPI (blue, inset; scale bar, 200 μm). Kaplan–Meier analysis and the log rank statistic (A): OX40$^{-/-}$ vs. WT, $P = 0.1410$; OX40L$^{-/-}$ vs. WT, $P = 0.0001$; OX40L$^{-/-}$ vs. OX40$^{-/-}$, $P = 0.0003$. Tukey's honestly significant difference test (B and D): OX40$^{-/-}$ vs. WT, $P = 0.0099$ (B); OX40L$^{-/-}$ vs. WT, $P = 0.0001$ (B); OX40L$^{-/-}$ vs. OX40$^{-/-}$, $P = 0.0039$ (B); WT donor, $P = 0.0109$ (D); OX40L$^{-/-}$ donor, $P = 0.0178$ (D).

Source data are available online for this figure.

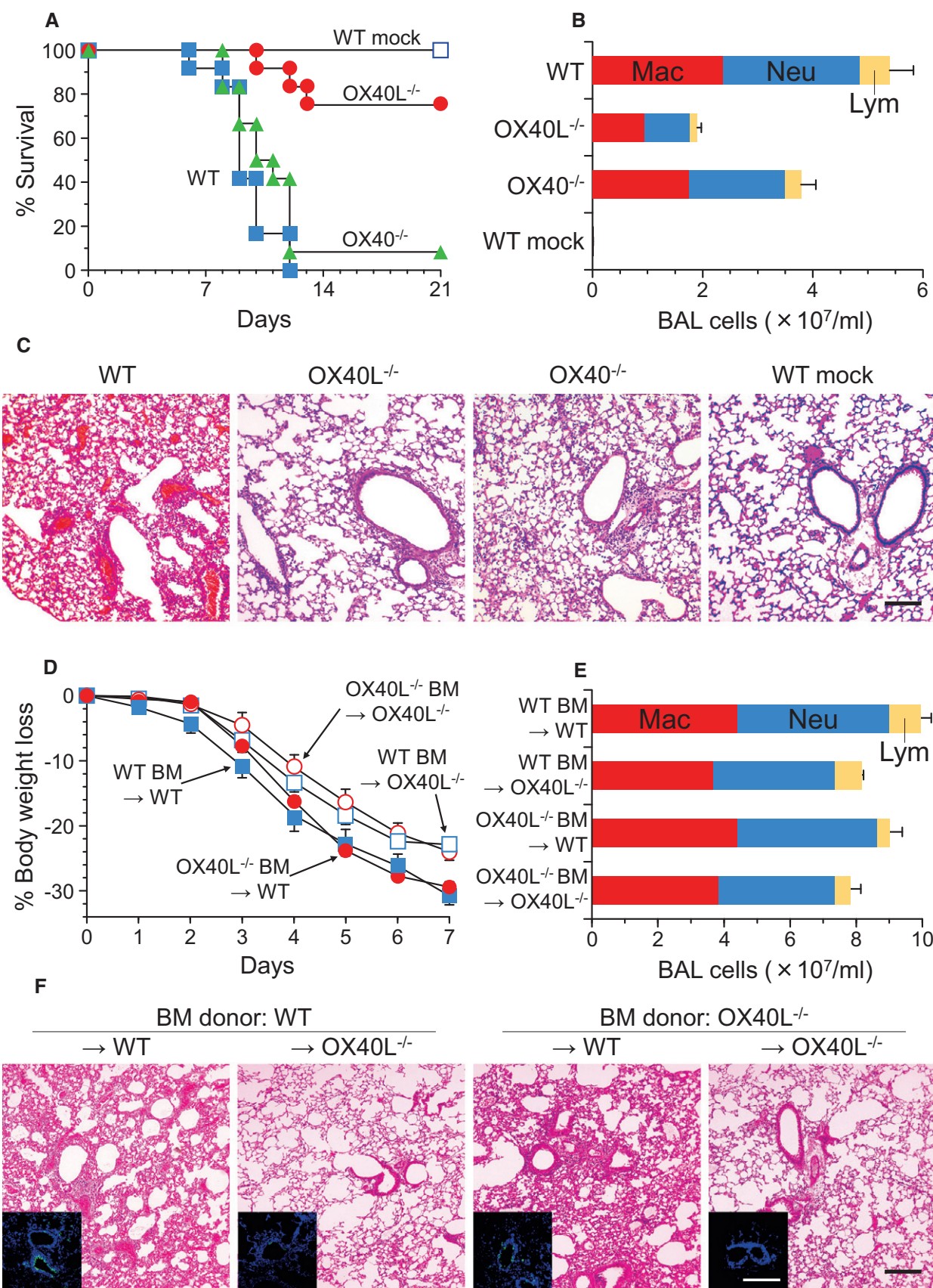

**Figure 1.**

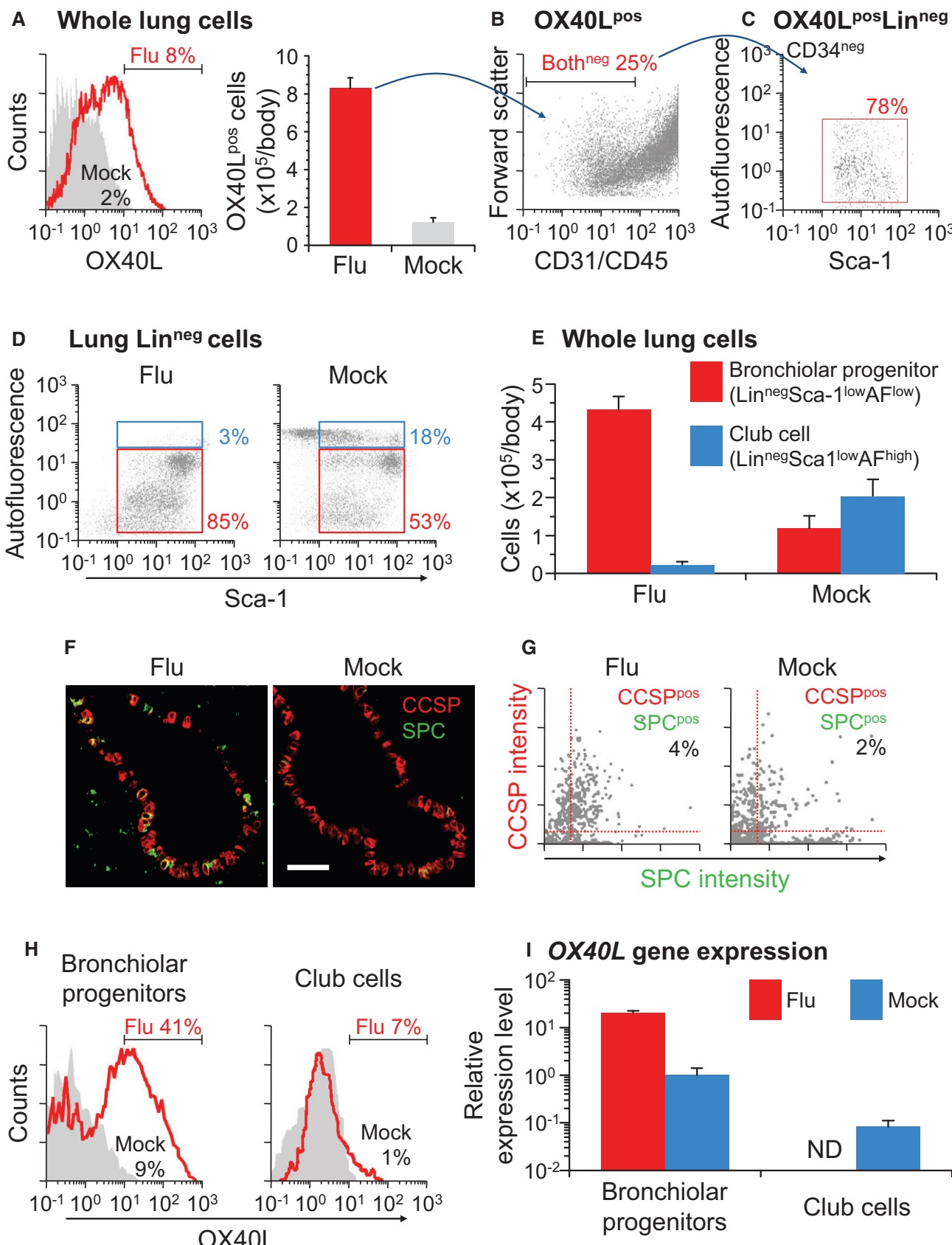

**Figure 2.**

◀ **Figure 2. Both the number and the OX40L expression level of bronchiolar progenitors were increased by influenza infection.**
Wild-type mice were intratracheally infected with a lethal dose of influenza A/H1N1 virus (Flu) or saline (Mock), and 7 days later, their lung cells and sections were evaluated.

A    Cell counts of OX40L-positive cells in whole lung cells.

B    Status of CD31 and CD45 expression in lung OX40L-positive cells.

C    Status of Sca-1 expression and autofluorescence in lung OX40L-positive Lin-negative (i.e., $OX40L^{pos}CD31^{neg}CD45^{neg}CD34^{neg}$) cells.

D, E  (D) Proportion of bronchiolar progenitors ($Lin^{neg}Sca-1^{low}AF^{low}$, red box) and club cells ($Lin^{neg}Sca-1^{low}AF^{high}$, blue box), and (E) their cell counts.

F, G  (F) Lung sections stained with antibodies to CCSP (red) and SPC (green), and (G) quantitative analysis of these imaging data. Scale bar, 50 μm.

H    Cell counts of OX40L-positive cells in bronchiolar progenitors and club cells.

I    *OX40L* gene expression in bronchiolar progenitors and club cells. By quantitative RT–PCR, the gene expression levels were analyzed relative to the bronchiolar progenitors of mock-infected mice. ND, not determined for scarcity of club cells after influenza infection.

Data information: Data are presented as the mean ± standard error of $n = 4$ (A and E) or $n = 3$ (I) per group. Student's unpaired two-tailed $t$-test (A and I): $P = 0.0001$ (A); progenitors, $P = 0.0102$ (I). Tukey's honestly significant difference test (E): progenitors, $P = 0.0001$; club cells, $P = 0.0097$.

Source data are available online for this figure.

## Increased number of bronchiolar progenitors

We then sought to determine whether the size of the bronchiolar progenitor population is increased in influenza-infected lungs (Fig 2D and E). Lung infection with influenza A/H1N1 virus increased the frequency of bronchiolar progenitors exhibiting a $Sca-1^{low}AF^{low}$ phenotype within the lung $Lin^{neg}$ fraction, but decreased that of club cells, formerly known as Clara cells, exhibiting a $Sca-1^{low}AF^{high}$ phenotype within the same fraction (Fig 2D). Concomitantly, influenza A/H1N1 infection caused a significant increase in the number of $Lin^{neg}Sca-1^{low}AF^{low}$ bronchiolar progenitors, and a significant decrease in that of $Lin^{neg}Sca-1^{low}AF^{high}$ club cells (progenitors, $P < 0.001$; club cells, $P < 0.01$; Fig 2E). Similar results were also achieved using influenza A/H3N2 virus (progenitors, $P < 0.001$; club cells, $P < 0.05$; Fig EV1A).

Because a CCSP and SPC dual-expressing phenotype has also been proposed as a characteristic of bronchiolar progenitors (Teisanu *et al*, 2009; Lau *et al*, 2012; Ardhanareeswaran & Mirotsou, 2013), we analyzed the CCSP/SPC dual expression by immunofluorescent staining of lung tissues from mice subjected to influenza A pulmonary infection (Fig 2F and G). The results showed that influenza A/H1N1 infection promoted the shedding of CCSP-immunoreactive club cells in terminal bronchioles and, instead, the appearance of CCSP/SPC dual-positive cells in such distal airways (Fig 2F). Correspondingly, the quantitative determination of the fluorescent signal intensity on the stained lung sections revealed that influenza A/H1N1 infection led to a twofold increase in the frequency of CCSP/SPC dual-positive cells as compared to mock infection (Fig 2G). Taken together, these data suggest that club cells in bronchioles are readily abolished by influenza A infection, whereas bronchiolar progenitors enlarge the population likely to repair the damaged epithelia.

## Influenza-induced OX40L expression on bronchiolar progenitors

To further understand the expression dynamics of OX40L on bronchiolar progenitors, we analyzed the OX40L expression by flow cytometry and quantitative RT–PCR studies (Fig 2H and I). As determined by flow cytometry, the frequency of bronchiolar progenitors expressing OX40L on the cell surface was increased up to 41% following influenza A/H1N1 infection, whereas that of club cells was increased only up to 7% (Fig 2H). Similar results were observed following influenza A/H3N2 infection (Fig EV1B). The quantitative RT–PCR analysis using *Gapdh* as the internal control

gene revealed the mRNA expression encoding OX40L to be significantly elevated in bronchiolar progenitors upon influenza A/H1N1 infection of the mouse airways ($P < 0.05$, Fig 2I). Similar results were achieved by using other internal control genes in the quantitative RT–PCR analysis (beta-2-microglobulin, $P < 0.05$; hypoxanthine phosphoribosyl-transferase 1, $P < 0.01$; Appendix Fig S4). These findings indicate that the pulmonary OX40L abundance associated with non-hematopoietic cells is augmented in influenza-infected mice as a result of both the enlarged population of bronchiolar progenitors and their increased expression of OX40L.

## OX40L-mediated susceptibility of bronchiolar progenitors to influenza A virus

To examine the functional implications of bronchiolar progenitors and their OX40L, we sorted bronchiolar progenitors and club cells, or $OX40L^{pos}$ and $OX40L^{neg}$ lung cells, after influenza A/H1N1 lung infection (Fig 3A–E). The expression of the mRNA encoding influenza nucleoprotein (*NP*) was assayed for the susceptibility to influenza viral infection by semiquantitative and quantitative RT–PCR. In wild-type mice, bronchiolar progenitors displayed a significantly higher level of the viral nucleoprotein expression than club cells ($P < 0.05$, Fig 3A). The nucleoprotein expression level was significantly attenuated in OX40L-deficient bronchiolar progenitors as compared with wild-type ones ($P < 0.05$, Fig 3B). When analyzing the differences between the $OX40L^{pos}$ and $OX40L^{neg}$ lung cells of influenza-infected wild-type mice, we found an approximately sixfold increase in the nucleoprotein expression in $OX40L^{pos}$ lung cells, which was consistent with the immunofluorescence studies showing matrix protein 2 (M2) of influenza virus only in $OX40L^{pos}$ lung cells ($P < 0.01$, Fig 3C and D). The compatibility between OX40L and influenza A/H1N1 virus was confirmed by OX40L/M2 dual immunostaining, in which OX40L and influenza matrix protein 2 tended to merge with each other, especially in bronchiolar progenitors (high and low magnification, Fig 3E and F). In aggregate, these data show that bronchiolar progenitors are more susceptible to influenza A virus, at least in part due to their OX40L expression, than club cells.

## Less productive replication of influenza virus in lung tissues of OX40L-deficient mice

Given that almost complete protection against otherwise lethal influenza pneumonia is achieved in mice lacking OX40L, which is

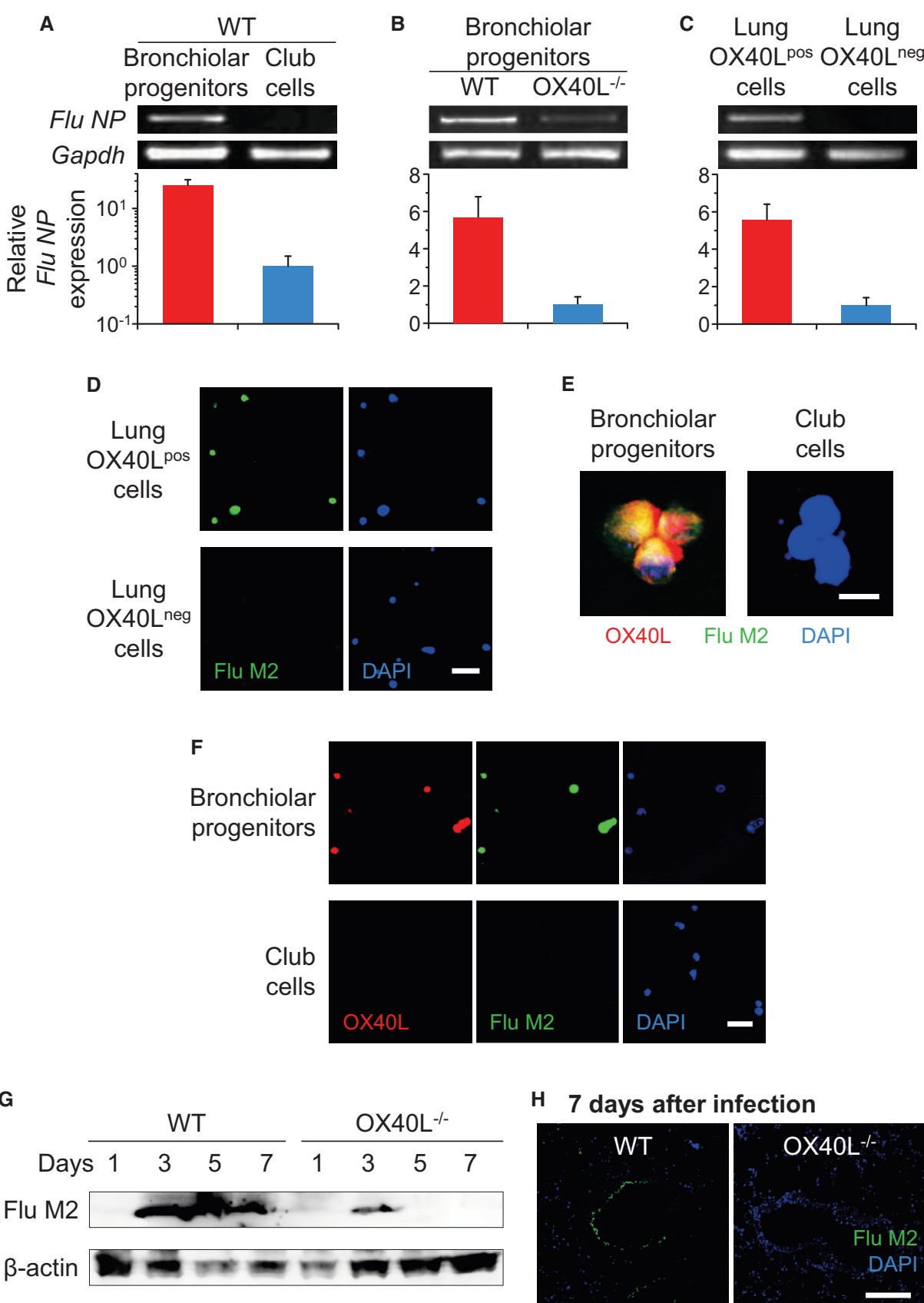

**Figure 3.**

**Figure 3. Bronchiolar progenitors are susceptible to influenza infection due to their OX40L expression.**

Wild-type and OX40L$^{-/-}$ mice were intratracheally infected with a lethal dose of influenza A/H1N1 virus, and their lung cells and sections were evaluated 3 days later except in panels (F–H).

A–C   By semiquantitative and quantitative RT–PCR, the levels of influenza virus nucleoprotein (*NP*) gene expression were analyzed in (A) wild-type bronchiolar progenitors relative to club cells, (B) wild-type bronchiolar progenitors relative to OX40L$^{-/-}$ ones, and (C) wild-type OX40-positive cells relative to OX40-negative cells. Mouse glyceraldehyde-3-phosphate dehydrogenase (*Gapdh*) mRNA expression was used as a control.

D   Cytospins prepared from sorted OX40L-positive and OX40L-negative lung cells of infected wild-type mice were immunostained with antibody to M2 protein of influenza virus, and the nuclei were identified with DAPI. Scale bar, 50 μm.

E   The study was similar to that in panel (D), but sorted bronchiolar progenitors and club cells from influenza-infected wild-type mice were stained with antibody to OX40L (red) as well as M2 protein (green). Scale bar, 10 μm.

F   This study was similar to that in panel (E), but bronchiolar progenitors and club cells were sorted at 7 days after the infection. Scale bar, 50 μm.

G   Representative Western blot of influenza virus M2 protein in lungs of wild-type and OX40L$^{-/-}$ mice on the indicated days after influenza infection. β-Actin was used as a loading control.

H   Lung sections from influenza-infected wild-type and OX40L$^{-/-}$ mice were immunostained with antibody to M2 protein of influenza virus, and the nuclei were identified with DAPI. Scale bar, 100 μm.

Data information: For panels (A–C), data are presented as the mean ± standard error of *n* = 3 per group. Student's unpaired two-tailed *t*-test (A–C): *P* = 0.0179 (A); *P* = 0.0173 (B); *P* = 0.0082 (C).

Source data are available online for this figure.

considered as a susceptibility factor to influenza A virus, we tested whether this characteristic of OX40L-deficient mice is associated with the viral replication in respiratory tissues (Fig 3G and H). To address this, we performed immunoblotting of whole-cell extracts from respiratory specimens of influenza-infected mice. The results revealed that influenza A/H1N1 virus-derived matrix protein 2 in wild-type mice markedly increased for 3 days post-infection and then decreased gradually, whereas that in OX40L-deficient mice similarly but modestly increased for 3 days and then decreased rapidly below detectable limits (Fig 3G). Consistently, at 7 days after the infection, viral matrix protein 2 was abundantly observed in bronchiolar and alveolar epithelial cells of wild-type mice in contrast to lung sections of OX40L-deficient mice where the viral protein was hardly observed (Fig 3H). The reduced viral replication in OX40L-deficient mice was also confirmed by lower numbers of influenza viral plaques in the BAL fluids (*P* < 0.5, Appendix Fig S5). These results strengthened our view that influenza-induced OX40L expression on bronchiolar progenitors promotes their susceptibility to influenza A virus, leading to viral growth in the infected lung.

## Basal status of the influenza virus receptor on bronchiolar progenitors

Human-adapted influenza A virus has been shown to initiate the infection via their hemagglutinin, which binds to α-2,6 sialic acid, sialic acid linked to galactose by α-2,6 linkage (Wilks *et al*, 2012). We next investigated whether the sialic acid modification status is relevant to the cellular susceptibility by fluorescent staining with α-2,6 sialic acid-specific lectin of *Sambucus nigra*. At the basal level, bronchiolar progenitors had higher levels of α-2,6 sialic acid on their cell surfaces as compared to club cells, suggesting a mechanistic link between the sialic acid modification and the cellular susceptibility to influenza A/H1N1 virus (Fig 4A). However, the abundance of α-2,6 sialic acid in bronchiolar progenitors was not simply a result of the sialylation-catalyzing activity, as analysis of our microarray data, whose details will be available in the ArrayExpress database (accession E-MTAB-4460), revealed that gene expressions for the responsible enzyme galactoside α-2,6-sialyltransferase (*St6gal1* and *St6gal2*) in bronchiolar progenitors were not higher but rather significantly lower than those in club cells (*St6gal1*, *P* < 0.01; *St6gal2*, *P* < 0.05; Fig 4B).

**Figure 4. Mouse OX40L mediates influenza infection *in vitro* and *in vivo*.**

A   α-2,6 sialic acid-specific *Sambucus nigra* lectin staining of bronchiolar progenitors and club cells from naive wild-type mice. Unstained cells were used as a control (Ctrl).

B   β-galactoside α-2,6-sialyltransferase (*St6gal1* and *St6gal2*) gene expression in bronchiolar progenitors and club cells from naive wild-type mice. By microarray analysis, the gene expression levels were analyzed and assessed as the Agilent gScale signal intensity.

C   Mouse OX40L-transfected MDCK cells were infected *in vitro* with influenza A/H1N1 virus. By semiquantitative and quantitative RT–PCR, the influenza virus *NP* gene expression levels were analyzed relative to those in OX40L-transfected cells 24 h after the infection. Endogenous canine *GAPDH* mRNA expression was used as a control.

D   The study was similar to that in panel (C), but MDCK cells were stained with antibody to mouse OX40L (red) and M2 protein (green). The nuclei were identified with DAPI. Scale bar, 50 μm.

E   Mouse OX40L-transfected MDCK cells were stained with α-2,6 sialic acid-specific *Sambucus nigra* lectin, and the mean fluorescent intensity was measured by flow cytometry. Controls included pNull-transfected MDCK cells.

F   The study was similar to that in panel (C), but the cells were pretreated with anti-mouse OX40L antibody for 24 h before the infection. The levels of influenza virus *NP* gene expression were analyzed relative to those in control antibody-pretreated cells.

G   Wild-type mice were intratracheally infected with a lethal dose of influenza A/H1N1 virus (day 0). On day 1 after the infection, mice were treated with intranasal administration of anti-mouse OX40L antibody or control antibody, and the susceptibility was determined by the survival of the mice (*n* = 10).

Data information: For panels (B, C and F), data are presented as the mean ± standard error of *n* = 3 per group. Student's unpaired two-tailed *t*-test (B, C, and F): *St6gal1*, *P* = 0.0058 (B); *St6gal2*, *P* = 0.0473 (B); *P* = 0.0012 (C); *P* = 0.0161 (F). Kaplan–Meier analysis and the log rank statistic (G): *P* = 0.0177.

Source data are available online for this figure.

   

**Mouse OX40L serves as a receptor for influenza infection**

Together, our results suggested that influenza A virus could attach to mouse OX40L via its α-2,6 sialic acid modification. To test this hypothesis, we constructed plasmid vectors expressing mouse

OX40L constitutively and transfected canine epithelial MDCK cells with them (Fig 4C–G). The transfected MDCK cells were infected with influenza A/H1N1 virus *in vitro*, and the expression derived from influenza-derived nucleoprotein was analyzed by semiquantitative and quantitative RT–PCR. Mouse OX40L-expressing MDCK

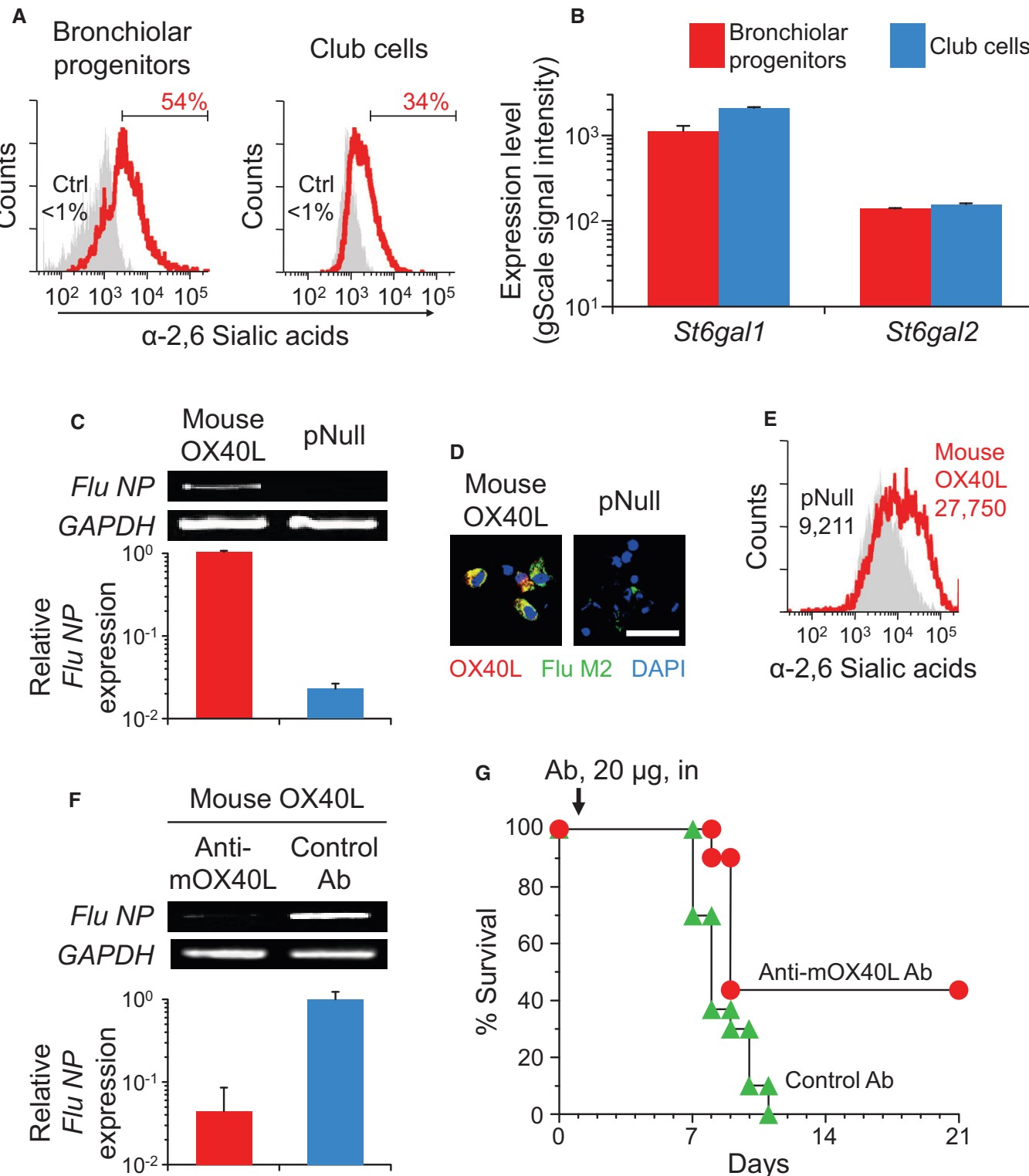

**Figure 4.**

cells expressed significantly higher levels of nucleoprotein than control cells, indicating that the forced mouse OX40L expression is functionally crucial for influenza infection ($P < 0.005$, Fig 4C). This was also demonstrated by the visualization of exogenous mouse OX40L protein expression that was well merged with matrix protein 2 of the influenza virus (Fig 4D). When staining transfected MDCK cells with fluorescently labeled lectin specific to α-2,6 sialic acid, the increase in α-2,6 sialic acid with OX40L-expressing cells suggested that exogenous mouse OX40L would be preferentially glycosylated and modified by α-2,6 sialic acid (Fig 4E). These observations were supported by prevention of the OX40L-mediated infection with sialidase treatment (Appendix Fig S6). The mouse OX40L-mediated influenza A viral infection was further confirmed by *in vitro* and *in vivo* blocking studies with the monoclonal antibody to mouse OX40L; the *in vitro* pretreatment of OX40L-expressing MDCK cells and the *in vivo* treatment of influenza-infected mice with anti-mouse OX40L antibody resulted in abrogated viral nucleoprotein expression and survival advantage, respectively ($P < 0.05$ to control antibody, Fig 4F; $P < 0.05$ to control antibody, Fig 4G).

### Human OX40L also serves as a receptor for influenza infection

Because mouse and human OX40L are type II glycoproteins with 46% identity and 67% similarity at the amino acid level, we next investigated whether human OX40L can be modified with α-2,6 sialic acid and be implicated in mediating the influenza A infection as a cell surface receptor.

Essentially similar results to those of mouse OX40L were achieved in MDCK cells expressing human OX40L. When infected with influenza A/H1N1 virus and A/H3N2 *in vitro*, human OX40L transfection significantly promoted the viral nucleoprotein expression, and the viral nucleoprotein expressions of either transfected MDCK cells were completely blunted by pretreatment with sialidase, which removes α-2,6 sialic acid from the cell surface (human OX40L compared to control pNull without the sialidase pretreatment, A/H1N1, $P < 0.01$, Fig 5A; A/H3N2, $P < 0.001$, Fig EV1C). Consistent with this finding that the OX40L-promoted influenza infection depended on the sialylation, forced expression of human OX40L led to a significant increase in not only the viral plaques, but also the surface sialic acid residues in α-2,6 linkage to the underlying sugar, the structure that enables the influenza A virus to attach and enter the host cells ($P < 0.01$, Appendix Fig S7; $P < 0.005$, Fig 5B and C). The expression levels of type I IFNs that potentially mount an antiviral response were comparable between naive and transfected MDCK cells (IFN-α, $P > 0.9$; IFN-β, $P > 0.9$; Appendix Fig S8).

We therefore sought to identify the glycosylation site responsible for the influenza infection in human OX40L (Fig 5D–F). Because 4 putative glycosylation sites were found in human OX40L, which is comprised of 183 amino acids, several plasmid vectors expressing mutated human OX40L were prepared and used for the transfection of MDCK cells (Fig 5D). There were no significant differences in the expression levels of wild-type and mutant genes of human OX40L ($P > 0.9$; Appendix Fig S9). Deletion of amino acids 121–183 and substitution of [114]asparagine had no impact on the OX40L-mediated effect to increase the influenza susceptibility, while deletion of amino acids 57-183 and substitution of [90]asparagine significantly

impaired the effect (compared to human OX40L; [121]Lys, $P > 0.9$; [57]Pro, $P < 0.05$; [114]Asn, $P > 0.9$; [90]Asn, $P < 0.05$; Fig 5E). We observed similar results for the α-2,6 sialic acid modification (compared to human OX40L; [121]Lys, $P > 0.5$; [57]Pro, $P < 0.005$; [114]Asn, $P > 0.3$; [90]Asn, $P < 0.005$; Fig 5F), clearly showing that [90]asparagine of human OX40L is indispensable for the receptor function in influenza infection.

From a therapeutic point of view to help defend against influenza A infection, we next screened for monoclonal antibody targeting [90]asparagine of human OX40L. Among 9 examined clones of anti-human OX40L antibody, only clone W66 had less affinity for the mutant human OX40L with alanine substituted for [90]asparagine ([90]Asn→Ala) than for the wild-type human OX40L (Fig EV2). Besides the point-mutated form [90]Asn→Ala, the W66 anti-human OX40L antibody showed a significantly reduced affinity for the deletion mutant [57]Pro→Stop as compared with the wild-type human OX40L, but did not show reduced affinities for other mutants including [121]Lys→Stop and [114]Asn→Ala ([121]Lys, $P > 0.05$; [57]Pro, $P < 0.001$; [114]Asn, $P > 0.9$; [90]Asn, $P < 0.001$; Fig 5G). We then investigated the functional role of the W66 anti-human OX40L antibody in shaping the *in vitro* infectious outcome of influenza A/H1N1 and A/H3N2 viruses in human OX40L-expressing MDCK cells (Figs 5H and I, and EV1D). Pretreatment with the W66 antibody markedly attenuated not only the RNA expression of the viral nucleoprotein but also the protein expression of the M2 proton channel, both of which are resultant products of influenza A infection (A/H1N1, $P < 0.05$ to control antibody, Fig 5H; A/H3N2, $P < 0.05$ to control antibody, Figs 5I and EV1D).

## Discussion

Here, we propose a novel mechanism by which influenza virus infection of the lower respiratory tract, which is clinically recognized as viral pneumonia, is exacerbated in paradoxical host reactions. In general, influenza infection is limited to the upper respiratory tract, where the epithelial environment is abundant in α-2,6 sialic acid receptors for human viral hemagglutinin (Wilks *et al*, 2012). Once the infection has reached the lower respiratory tract and produced epithelial damage, bronchiolar progenitors increase in number to replenish damaged epithelial cells, and enhance the OX40L expression on their surface to bolster the host immune response for virus elimination. However, these host defense reactions in turn worsen the influenza pneumonia by enhancing the viral binding to the airway epithelium through the newly emerging α-2,6 sialic acid of glycosylated OX40L protein expressed on expanded bronchiolar progenitors.

Influenza is generally a self-limiting upper respiratory infection, commonly referred to as flu, which sometimes results in pneumonia. Influenza pneumonia is a significant cause of morbidity and mortality (Kuiken *et al*, 2012). During the swine flu pandemic of H1N1 type A viruses in 2009, the rate of pneumonia was reported as 0.4% and the resulting mortality reached 25% (Cesario, 2012). The pathophysiology of some patients with a fatal outcome from influenza pneumonia is characterized by elevated levels of proinflammatory cytokines such as IL-6, tumor necrosis factor-α, and IFN-γ, sometimes referred to as the cytokine storm (To *et al*, 2010; Barik, 2012). While these proinflammatory cytokines are required

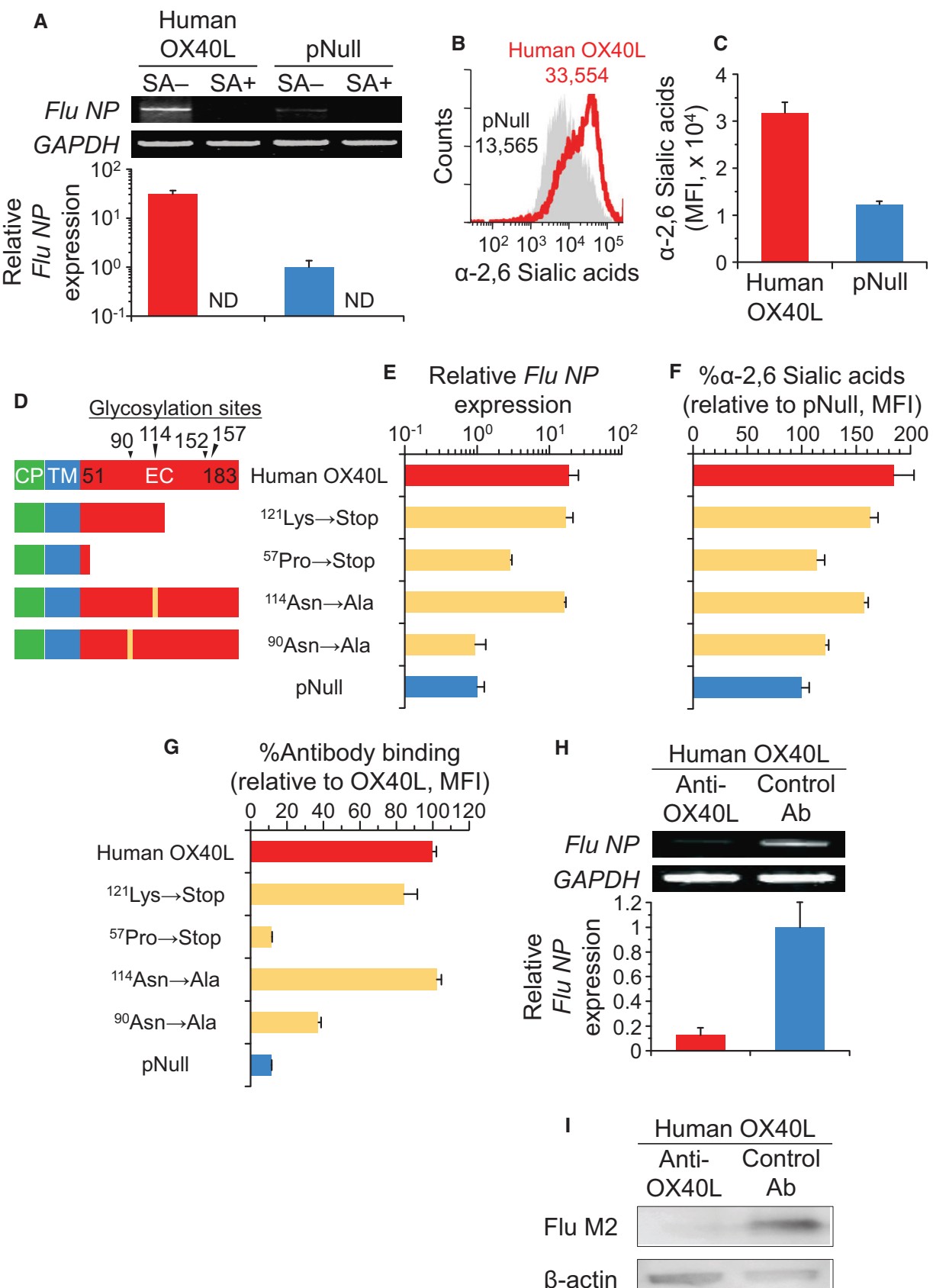

**Figure 5.**

◀

**Figure 5. Human OX40L mediates influenza infection *in vitro* and *in vivo*.**

A   Human OX40L-transfected MDCK cells were pretreated with or without sialidase for 24 h. After the sialidase treatment, the cells were infected *in vitro* with influenza A/H1N1 virus, and 24 h later, the levels of influenza virus *NP* gene expression were analyzed by semiquantitative and quantitative RT–PCR. The intensity was quantified relative to pNull-transfected cells. Endogenous canine *GAPDH* mRNA expression was used as a control. SA, sialidase; ND, not detectable.

B, C   Human OX40L-transfected MDCK cells were stained with α-2,6 sialic acid-specific *Sambucus nigra* lectin, and the mean fluorescent intensity (MFI) was measured by flow cytometry.

D   Cartoon depicting wild-type and mutant constructs of human OX40L gene. CP, cytoplasmic domain; TM, transmembrane domain; EC, extracellular domain.

E, F   The study was similar to that in panels (A and C), respectively, but MDCK cells were transfected with mutant human OX40L genes as well as the wild-type one.

G   The transfected MDCK cells were stained with W66 anti-human OX40L antibody, and the MFI was measured by flow cytometry.

H   Before *in vitro* infection with influenza A/H1N1 virus, OX40L-transfected MDCK cells were pretreated with W66 anti-human OX40L or control antibodies for 24 h. As in panel (A), the influenza virus *NP* gene expression levels were analyzed relative to those in control antibody-pretreated cells.

I   The study was similar to that in panel (H), but influenza virus M2 protein was evaluated by Western blot analysis using β-actin as a loading control. Scale bar, 100 μm.

Data information: Data are presented as the mean ± standard error of $n = 3$ per group except for panels (B, D and I). Student's unpaired two-tailed *t*-test (A, C, and H): OX40L vs. pNull, no sialidase, $P = 0.0069$ (A); $P = 0.0011$ (C); $P = 0.0142$ (H). Tukey's honestly significant difference test compared to OX40L (E–G): [121]Lys, $P = 0.9990$ (E); [57]Pro, $P = 0.0369$ (E); [114]Asn, $P = 0.9911$ (E); [90]Asn, $P = 0.0175$ (E); [121]Lys, $P = 0.5702$ (F); [57]Pro, $P = 0.0016$ (F); [114]Asn, $P = 0.3560$ (F); [90]Asn, $P = 0.0041$ (F); [121]Lys, $P = 0.0505$ (G); [57]Pro, $P = 0.0001$ (G); [114]Asn, $P = 0.9959$ (G); [90]Asn, $P = 0.0001$ (G).

Source data are available online for this figure.

---

for cytotoxic mechanisms to destroy infected cells and clear the virus, excessive levels give rise to systemic inflammatory response syndrome, which is known to cause multiorgan dysfunction, consisting of destruction and desquamation of airway epithelium, inflammatory cell infiltration around the airways, and diffuse alveolar damage (Peiris *et al*, 2010; Barik, 2012; Damjanovic *et al*, 2012).

For the ensuing repair and regeneration of the injured lung architecture, lung stem and progenitor cell candidates have been proposed over the past several years. Stem cells are generally defined as rare cells with a capacity for both long-term self-renewal and multipotent differentiation. In contrast, progenitors are usually considered to be tissue-resident cells that proliferate infrequently in the steady state but can enter a continuous proliferation state after injury, and that can facilitate the transition between a differentiated state and a more differentiated state according to the injury/repair conditions (Stripp, 2008; Lau *et al*, 2012; Ardhanareeswaran & Mirotsou, 2013). Although whether adult lung stem cells exist or not still remains an open question, the concept that progenitor populations are responsible for normal lung epithelial maintenance has become largely accepted due to findings from animal models (Lau *et al*, 2012; Rackley & Stripp, 2012; Ardhanareeswaran & Mirotsou, 2013). In this context, our bone marrow chimera experiment indicated that OX40L on non-hematopoietic cells is implicated in exacerbating influenza pneumonia, albeit under experimental conditions of the irradiated recipients and the reconstructed immune system. Further experiment of flow cytometry and immunohistochemistry identified bronchiolar progenitors found by Stripp and colleagues as the OX40L[pos] non-hematopoietic cells that not only increase in number but also induce OX40L expression on their surface upon respiratory infection with influenza A virus (Teisanu *et al*, 2009, 2011).

Bronchiolar progenitors can be isolated based on flow cytometric isolation of lung cells exhibiting the Lin[neg]Sca-1[low]AF[low] phenotype (Teisanu *et al*, 2009, 2011; Lau *et al*, 2012). As demonstrated by functional assays, bronchiolar progenitors appear to be mainly composed of a special subset of club cells known as variant club cells, which reside in the distal airways and, after injury, rapidly reconstitute both secretory (club cell) and ciliated populations of the injured airway (Hong *et al*, 2001; Giangreco *et al*, 2002; Teisanu *et al*, 2011; Kotton, 2012). In support of these findings, a study observed that, at early time points after an influenza infection, the infected cells were phenotyped for bronchiolar progenitors (i.e., CCSP/SPC dual-positive), which might differentiate to club cells afterward, in the airways of the respiratory tract (Heaton *et al*, 2014). The cellular profile of bronchiolar progenitors is further supported by our *in vivo* results demonstrating that bronchiolar progenitors promptly enlarged their population after lethal influenza A/H1N1 and A/H3N2 infection, likely to regenerate the injured distal epithelia. Using a similar but sublethal A/H1N1 influenza infection, a remarkable subset of basal cells that commonly reside in the proximal airways has recently been reported to proliferate after the injury and differentiate into an alveolar lineage in the distal lung for the regeneration process (Kumar *et al*, 2011; Kotton, 2012). Although there may be a little overlap between the two progenitor subsets identified by us and others, these discordant findings highlight the further challenges to uncover the precise mechanisms by which progenitor populations in the lung undergo airway and alveolar differentiation.

Not only did this study show that the expansion of bronchiolar progenitors is triggered for the epithelial repair of influenza-damaged lungs, but it also revealed a link between the unexpected expression of OX40L and its contribution to the development of severe disease during viral infection. Costimulatory receptor–ligand interactions including OX40–OX40L have been intensively studied in life-threatening influenza immunopathology, because the detrimental outcome occurs mostly as a result of excessive inflammatory responses (Kopf *et al*, 1999; Humphreys *et al*, 2003, 2006; Lin *et al*, 2009; Snell *et al*, 2010; Braciale *et al*, 2012). In line with previous observations showing that OX40 blockade with soluble OX40–immunoglobulin fusion proteins improved influenza-driven illness, we also confirmed that inflammatory infiltrates were ameliorated to some extent in the influenza-infected lungs of OX40-deficient mice (Humphreys *et al*, 2003). Furthermore, we found that mice genetically deficient in OX40L are markedly protected from the lethal influenza virus infection owing to the viral hemagglutinin binding property of OX40L through the α-2,6 sialic acid (Wilks *et al*, 2012). Earlier studies on the role of the α-2,6 sialic acid during influenza virus infection in mice have shown the possibility that the virus enters cells using the α-2,3 sialic acid or an unknown receptor as well as the α-2,6 sialic acid (Glaser *et al*, 2007). Alternative moieties of OX40L may be also involved in its binding property to influenza virus.

Collectively, our finding that influenza-induced OX40L expression on expanded bronchiolar progenitors for antiviral host defense paradoxically exacerbates the influenza virus infection may provide key insights for an innovative therapeutic strategy for severe influenza infection.

# Materials and Methods

### Mice

OX40L-deficient and OX40-deficient mice were generated as described previously and had been backcrossed more than six generations to C57BL/6 mice (Pippig *et al*, 1999; Murata *et al*, 2000). As the wild-type control, C57BL/6 mice were purchased from Japan Charles River (Yokohama, Japan). For bone marrow transplantation, recipient mice were irradiated with 10 Gy and, on the following day, $3 \times 10^6$ adult bone marrow cells extracted from the femurs and tibias of the donor mice were transplanted into the 20-week-old recipients via the tail vein (Sun *et al*, 2013). Unless otherwise noted, all animals used in this study were randomized and matched for age (6–10 weeks old, unless otherwise noted), sex (female), and strain (C57BL/6 background) within each experiment and were housed under specific pathogen-free conditions. All procedures were performed without blinding, according to protocols approved by Tohoku University's Institutional Committee for the Use and Care of Laboratory Animals.

### Influenza virus infection

A murine-adapted influenza A/H1N1 virus (Puerto Rico/8/34, PR8) and an influenza A/H3N2 virus (A/Aichi/2/68) were prepared as previously described (Tanimoto *et al*, 2005; Yano *et al*, 2009). We intratracheally administrated $2 \times 10^4$ plaque-forming units (pfu, 5 times the minimal lethal dose, 20 µl) of influenza virus into anesthetized mice. For neutralizing the OX40L *in vivo*, 1 day later, we treated the infected mice with an intranasal administration of 20 µg anti-mouse OX40L antibody, clone MGP34 (Murata *et al*, 2000). Unless otherwise indicated, 7 days after the infection, bronchoalveolar lavage (BAL) fluids and lungs were obtained from the mice as described previously (Damayanti *et al*, 2010; Daito *et al*, 2011). Briefly, the lungs were lavaged twice with 0.75 ml of phosphate-buffered saline, pH 7 (PBS). After centrifugation, total BAL cells resuspended in 1 ml of PBS were counted with a hemocytometer. The cytospins were stained with Diff-Quik (Sysmex, Kobe, Japan) to differentiate macrophages, lymphocytes, neutrophils, and eosinophils on the basis of cellular morphology and staining characteristics. To isolate lung cells, exsanguinated lungs were incubated with airway-instilled elastase (4 units/ml, Worthington Biochemical, Lakewood, NJ) at 37°C for 15 min (0.5 ml, 3 times at 5-min interval) and were further treated with 1 ml of DNase I (25 µg/ml, Roche Applied Science, Indianapolis, IN) at 37°C for 10 min. From BAL cells and lung cells, red blood cells were excluded by using ACK lysing buffer (Life Technologies, Grand Island, NY). For histopathological examination of lung tissue, paraffin sections (5 µm thick) were fixed with 10% formaldehyde and were stained with hematoxylin and eosin (H&E).

### Flow cytometry

Isolated cells were resuspended as whole lung cells in 70 µl of Hank's balanced salt solution (Life Technologies) supplemented with 10 mM HEPES (4-2-hydroxyethyl-1-piperazineethanesulfonic acid, Life Technologies) and 2% fetal bovine serum (Nichirei, Tokyo, Japan) and were incubated with the following fluorescence-conjugated monoclonal antibodies for 45 min at 4°C in the dark: phycoerythrin (PE)-cyanine 7 (Cy7) anti-CD31 (clone 390, 1:40, eBioscience, San Diego, CA), PE-Cy7 anti-CD45 (clone 30F-11, 1:40, BD Biosciences, San Jose, CA), Alexa Fluor 647 anti-CD34 (clone RAM34, 1:40, eBioscience), and PE anti-Sca-1 (clone D7, 1:40, BD Biosciences) as described previously (Teisanu *et al*, 2009). To detect α-2,6 sialic acids or mouse OX40L expression, cells were stained with 100 µg/ml of fluorescein isothiocyanate (FITC)-conjugated *Sambucus nigra* agglutinin I and II (SNA, EY Laboratories, San Mateo, CA) or FITC-conjugated anti-mouse OX40L monoclonal antibody (clone OX-89, 1:40) for 1 h at 4°C (Shibuya *et al*, 1987). The binding affinities of anti-human OX40L antibodies (clones W9-1, W18, W66, 5A8, TAG-34, 8F4, TARM-34, HD-1, and HD-2) were evaluated by indirect staining with FITC-conjugated goat-antibody to rat IgG (for W9-1, W18, and W66; Abcam, Cambridge, MA), mouse IgG (for 5A8, TAG-34, 8F4, HD-1, and HD-2; Abcam), or mouse IgM (for TARM-34; Abcam) for 1 h at 4°C (Imura *et al*, 1996; Baba *et al*, 2001; Kasahara *et al*, 2013). Stained cells were analyzed and sorted using a FACSAria II cell sorter and a FACSDiva software (BD Biosciences). A viability dye 7-amino-actinomycin D (7-AAD, Beckman Coulter, Brea, CA) was used to eliminate dead cells.

### Cell line and the culture condition

The canine kidney epithelial cell line MDCK was obtained from the RIKEN BioResource Center (Tsukuba, Japan) and maintained at 37°C in 5% $CO_2$ with Dulbecco's modified Eagle medium (DMEM, Sigma-Aldrich, St. Louis, MO) supplemented with 10% fetal bovine serum (Nichirei) and 100 units/ml penicillin–streptomycin (Life Technologies). For plasmid transfection, $7 \times 10^4$ MDCK cells were seeded in a 12-well plate (BD Biosciences) and, 48 h later, were transfected with 1 µg of the plasmid DNA by Lipofectamine plus reagent (Life Technologies) following the manufacturer's instructions. Two days after the transfection, the transfected MDCK cells were washed twice with minimum essential medium (MEM, Life Technologies), and infected with influenza PR8 virus (H1N1) or A/Aichi/2/68 virus (H3N2) at a multiplicity of infection (MOI) of 0.001 in 100 µl of MEM for 90 min at 34°C with intermittent agitation to ensure uniform infection. After infection, the unadsorbed viruses were washed away twice with 1 ml of MEM, and the cell culture was maintained for 24 h at 34°C in 1 ml of MEM supplemented with 2 mM L-glutamine (Nissui Pharmaceutical, Tokyo, Japan), 25 mM HEPES, 0.4% bovine serum albumin (Wako Pure Chemical, Osaka, Japan), 0.1% DEAE-dextran hydrochloride (Sigma-Aldrich), and 2 µg/ml TPCK-trypsin (Takara Bio, Otsu, Japan), as described previously (Hashem *et al*, 2009). Where indicated, before the infection, the transfected MDCK cells were pretreated with 10 µunits/ml sialidase (from *Arthrobacter ureafaciens*, Roche Applied Science), 10 µg/ml anti-mouse OX40L antibody (clone MGP34), or anti-human OX40L antibody (clone W66) for 24 h.

**The paper explained**

**Problem**

Excessive immune responses are recognized as a pathophysiological feature of fatal influenza pneumonia. Costimulatory signals, such as OX40–OX40 ligand (OX40L) interactions, have been studied in the context of the hyperactive inflammation.

**Results**

A genetic deficiency of OX40L, but not OX40, enabled mice to survive otherwise lethal intratracheal challenge with influenza A virus. The protection depended mainly on OX40L of non-hematopoietic cells, which were characterized as bronchiolar progenitors. Upon infection, the bronchiolar progenitors were found to increase in number for repairing the influenza-damaged epithelia and to enhance the OX40L expression on their surfaces for attaining the antiviral immunity. However, these defense-like host responses result in more severe respiratory disease, because the induced OX40L is accompanied by the emergence of sialic acids with α-2,6 linkages, which bind to the viral hemagglutinin and provoke more extensive infection. The specific antibody targeting the sialylated site of OX40L substantially diminished the OX40L-mediated susceptibility to influenza.

**Impact**

The paper provides a hitherto undefined pathogenic role of OX40L, showing that influenza-induced OX40L expression on expanded bronchiolar progenitors for antiviral host defense paradoxically exacerbates the influenza infection by enhancing the viral binding to the progenitors. The increased affinity is due to the newly emerging α-2,6 sialic acid of glycosylated OX40L and should be considered as a potential therapeutic target in severe influenza infection.

## Plasmid vector construction

The plasmids, including pmOX40L, phOX40L, and pNull, are pBluescript II KS(+) vectors (Agilent Technologies, Santa Clara, CA) in which the mouse OX40L cDNA, human OX40L cDNA, and no transgene, respectively, are under transcriptional control of the cytomegalovirus (CMV) immediate-early enhancer and promoter. To modify the nucleic acid sequence of human OX40L in phOX40L, a PrimeSTAR mutagenesis kit (Takara Bio) was used according to the manufacturer's instructions. Amplification conditions were 30 cycles of 98°C for 10 s, 55°C for 15 s (58°C in marking a point mutation), and 72°C for 20 s. The modified nucleic acid sequence was confirmed by dye-terminator cycle sequencing with Applied Biosystems 3500xL Genetic Analyzer (Life Technologies).

## RNA assessments

Total cellular RNA was extracted from lung cells and MDCK cells by an RNeasy Plus kit (QIAGEN, Valencia, CA). For microarray analysis, we synthesized cRNA from the total cellular RNA samples with a Low Input Quick Amp Labeling kit for one color, hybridized it to SurePrint G3 Mouse GE 8 × 60K Microarray kit, and quantified the data as gScale signal intensities using a Feature Extraction software (all from Agilent Technologies). For reverse transcriptase–polymerase chain reaction (RT–PCR), total cellular RNA was reverse-transcribed to cDNA by a High Capacity RNA-to-cDNA kit (Life Technologies). The generated cDNA was amplified by semiquantitative and quantitative PCR. Semiquantitative PCR was conducted using Platinum *Taq* DNA polymerase (Life Technologies), and the

amplified products were resolved on a 2% agarose gel and detected by ethidium bromide staining; the semiquantitative amplification conditions were 94°C for 2 min, then 30 cycles of 94°C for 30 s, 58°C for 30 s, and 72°C for 45 s. Quantitative PCR was performed by a SYBR GreenER qPCR SuperMix Universal kit (Life Technologies) in the DNA Engine Opticon 2 system (Bio-Rad Laboratories, Hercules, CA) as suggested by the manufacturer. The quantitative data were normalized to the mouse or canine glyceraldehydes-3-phosphate dehydrogenase (mouse *Gapdh* or canine *GAPDH*) expression unless otherwise noted, and the relative gene expression was determined as a factor by which the normalized expression of the sample was changed from that of the reference (Wu *et al*, 2010; Kanehira *et al*, 2012). The primer pairs used in this study were as follows: influenza nucleoprotein (*Flu NP*), 5′-ACTCACATGATGATCTGG-3′ and 5′-CTGCATTGTCTCCGA AGA-3′; mouse *OX40L* 5′-CCCTCCAATCCAAAGACTCA-3′ and 5′-AT CCTTCGACCATCGTTCAG-3′; mouse *Gapdh*, 5′-TGTGTCCGTCGT GGATCTGA-3′ and 5′-CCTGCTTCACCACCTTCTTGA-3′; and canine *GAPDH*, 5′-TGAAGGTCGGAGTCAACGGATTTGGT-3′ and 5′-CAT GTGGGCCATGAGGTCCAC-3′.

## Immunofluorescence

For immunocytochemistry, $2 \times 10^4$ cells were cytospun onto glass slides ($600 \times g$, 2 min) and were fixed in 10% neutral buffered formalin for 20 min. After being treated with Protein Block Serum-Free (Dako, Carpinteria, CA) for 30 min, the cytospun cells were incubated with primary monoclonal antibodies to influenza A virus M2 protein (clone 14C2, 1:500, Abcam) and to mouse OX40L (clone RM134L, 1:500, eBioscience) for overnight at 4°C. For immunofluorescent staining of lung tissues, 5-μm cryosections were dehydrated in 100% ethanol and rehydrated in decreasing concentrations of ethanol in PBS. When necessary, antigen retrieval was performed by incubation in water-diluted Histofine (pH 9, Nichirei) and treatment with an autoclave (15 min, 121°C). After the Protein Block Serum-Free treatment, the following primary antibodies were added and incubated for overnight at 4°C: anti-CCSP (club cell secretory protein, 1:500, Santa Cruz Biotechnology, Dallas, TX) and anti-SPC (surfactant protein C, 1:500, Santa Cruz Biotechnology), or anti-influenza A virus M2 protein (clone 14C2, 1:500, Abcam). Slides were treated with a fluorescence-labeled secondary antibody (1:200, Life Technologies) for 1 h at 25°C and were mounted using VECTASHIELD Mounting Medium with DAPI (4′,6-diamidino-2-phenylindole, Vector Laboratories, Burlingame, CA). Fluorescent images were acquired by using an LSM 780 confocal microscope (Carl Zeiss, Oberkochen, Germany) and were analyzed by 2 independent investigators (Kikuchi and Tode). Where indicated, the images were quantified by using a TissueFAXS system (TissueGnostics, Vienna, Austria).

## Western blotting

Lung cells or MDCK cells were lysed in RIPA buffer (Cell Signaling Technology, Danvers, MA) containing Protease Inhibitor Cocktail (Sigma-Aldrich). An aliquot of the lysate (50 μg for lung cells or 20 μg for MDCK cells) was separated in a 10% Bis-Tris gel (Life Technologies) and transferred onto a polyvinylidene difluoride membrane (PVDF, Life Technologies) by using Trans-Blot semi-dry

electrophoretic transfer system (Bio-Rad Laboratories). After treatment with PVDF Blocking Reagent (TOYOBO, Osaka, Japan), the membrane was probed with a primary monoclonal antibody against influenza A virus M2 protein (clone 14C2, 1:500, Abcam) or β-actin (clone AC-15, 1:2000, Sigma-Aldrich) and a horseradish peroxidase (HRP)-conjugated secondary antibody against the primary antibody (1:500, Santa Cruz Biotechnology). The signals were visualized using an ECL Western Blotting Detection Reagents (GE Healthcare, Piscataway, NJ).

### Statistical analysis

Two data sets were compared by Student's unpaired two-tailed *t*-test. For multiple comparisons, *post hoc* analysis was performed by Tukey's honestly significant difference test. In survival studies, mortality was compared with Kaplan–Meier analysis and the log rank statistic. At least three samples or animals in each group were used for statistical analysis. Normal distribution of the data and the equal variances were evaluated by histograms and F-test statistics. $P$-values $< 0.05$ were considered statistically significant. Error bars in the graphical data represent the means $\pm$ standard error.

**Expanded View** for this article is available online.

### Acknowledgements
We thank Mitsu Takahashi for her technical assistance and Brent Bell for reading the manuscript. These studies were supported, in part, by Grants-in-Aid for Scientific Research from the Ministry of Education, Culture, Sports, Science and Technology (Nos. 23390219 and 15K15316, Tokyo, Japan) and the Core Research for Evolutional Science and Technology Program from the Japan Science and Technology Agency (Tokyo, Japan).

### Author contributions
TH contributed to conception and design and acquisition of data; TK contributed to conception and design, data analysis and interpretation, and manuscript writing; NT, AS, MY, YM, and RK contributed to data analysis and interpretation; TK, TT, NI, YT, and HN contributed to conception and design; TN, AW, and MI contributed to conception and design and administrative support.

### Conflict of interest
The authors declare that they have no conflict of interest.

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
