## [Review Process File · EMBO Molecular Medicine]

OX40 ligand newly expressed on bronchiolar progenitors mediates influenza infection and further exacerbates pneumonia

Taizou Hirano, Toshiaki Kikuchi, Naoki Tode, Arif Santoso, Mitsuhiro Yamada, Yoshiya Mitsuhashi, Riyo Komatsu, Takeshi Kawabe, Takeshi Tanimoto, Naoto Ishii, Yuetsu Tanaka, Hidekazu Nishimura, Toshihiro Nukiwa, Akira Watanabe, and Masakazu Ichinose

Corresponding author: Toshiaki Kikuchi, Niigata University Graduate School of Medical and Dental Sciences

Review timeline:

Submission date:	15 January 2015
Editorial Decision:	01 February 2015
Resubmission:	16 December 2015
Editorial Decision:	11 January 2016
Revision received:	11 February 2016
Accepted:	16 February 2016

Transaction Report:

Editor: Céline Carret

1st Editorial Decision

01 February 2015

Thank you for the submission of your manuscript to EMBO Molecular Medicine. We have now heard back from the two referees whom we asked to evaluate your manuscript. Although the referees find the study to be of potential interest, they also raise a number of concerns that must be addressed in the next final version of your article.

As you will see from the comments below, both referees find the study interesting and novel. However, they both recommend performing additional experiments and replicating some of the key data to improve and strengthen the findings. As the reports are very clear and explicit I will not get into further details, but please note that it is EMBO Molecular Medicine policy to allow only a single round of revision and that, as acceptance or rejection of the manuscript will depend on another round of review, your responses should be as complete as possible.

Please see instructions below for submitting your revised article. Please do not forget to include the authors' checklist and make sure that all requested information can be found within the main text.

I look forward to seeing a revised form of your manuscript as soon as possible.

***** Reviewer's comments *****

Referee #1 (Comments on Novelty/Model System):

Some additional experiments, and repeating some experiments, as specified in the remarks to the author, should enhance the conclusions in the paper and make it more convincing.

Referee #1 (Remarks):

This is a very interesting paper, suggesting that OX40L can be a receptor for influenza virus and aid infection of bronchial epithelial cells. In general the experiments are quite convincing but additional experiments are needed to solidify and enhance the conclusions.

1. In Figure 1a, the authors suggest that OX40L^{-/-} mice are protected from death with high dose intratracheal PR8/H1N1 infection whereas OX40^{-/-} mice are not. This is perhaps the most important data as it leads the rest of the paper in supporting the notion that the any results on lung pathology are not simply a repeat of what has been published before, which claimed that OX40-OX40L interactions drive an exacerbated T cell response to flu virus that causes pathology (Humphreys et al, JEM, 2003). Although the data as presented look convincing, this experiment was apparently only 1 experiment with 12 mice per group, and it is not indicated how survival was measured (presumably a loss of a certain amount of body weight given animal welfare rules). There are some differences in BAL counts and histology that somewhat correlate with the survival curves, but these are not so different to be convincing that this would fully explain the survival results. In Fig. 1d body weight loss in BM chimeras is shown, with very little change between groups, again questioning how real the data are in Fig. 1a. Thus, the contention that OX40^{-/-} mice differ from OX40L^{-/-} mice needs to be substantiated, with more repeat experiments, showing weight loss curves as well as survival curves. Importantly this should be performed with different doses of the virus to show the result does not simply apply to one specific situation.

2. The authors suggest that OX40L is induced on bronchiolar progenitors by both H1N1 infection (Fig. 2H) and H3N2 infection (Fig. 3B) but the latter results are not convincing that OX40L is significantly or strongly upregulated. Again, as the conclusion of the paper hinges on OX40L being active on these cells independent of OX40, experiments should also be performed assessing weight loss and survival in OX40L^{-/-} and OX40^{-/-} with H3N2 to show that the results are not simply specific for one strain of flu.

3. The authors imply that OX40L^{-/-} mice are protected from lethal flu infection in part because bronchiolar progenitor cells cannot be infected equivalently due to OX40L directly binding the virus. They show M2 expression in WT vs OX40L^{-/-} mice (Fig. 4G). If this is significant, and again not related to an immune response involving OX40-OX40L interactions, a kinetic analysis of levels of total virus particles (conventional plaque assay) as well as M2 protein in the lungs of OX40L^{-/-} and OX40^{-/-} mice over the first week of infection should also be dramatically different. Performing the experiment with H1N1 and H3N2 would again substantiate the conclusions.

4. There is a disconnect in the relative NP expression in mouse OX40L transfected cells in Fig. 6A vs 6D. Can the authors explain why the qPCR results are substantially different from the semi-qPCR.

5. In Fig. 7, the authors present data with human OX40L transfected cells treated with sialidase. It would also be useful to perform the same study with mouse OX40L.

6. The authors need to control for OX40L transfection in MDCK cells. Based on data in Fig. 8, it is not clear how uniform OX40L is expressed and the implication is that only a subset of cells may have expressed high amounts. This could impact the viral infection data. Also the data with the mutant OX40L constructs can only be evaluated effectively if similar levels of OX40L are expressed on the cell surface with each mutant. What are the absolute levels of OX40L on transfected cells and how do they compare to bronchiolar progenitors.

7. The final data staining OX40L in tissue sections from autopsy are not convincing. The data are

not clear and it is impossible to see any real staining. Moreover, bronchial epithelial cells are notoriously known to stain with almost any antibody. There is no control for the antibody to show it stains anything specifically. The comparison between tissues is also not valid, and making conclusions from tissues from single patients whose histories are not documented also calls into question any conclusions drawn.

Minor point:

Some of the data are not logically placed. e.g. the first data on sialic acids (Fig. 4H and I) would be better in the current Fig. 6. There is no need for a separate Fig. 5 as this data is related to that in Fig. 4. Similarly, data in Fig. 3C and D are not mentioned until Fig. 7 is discussed and would be better in Fig. 7. Also, data in Fig. 8 should either be in a supplementary Figure, or data in Fig. 8 combined with Fig. 7G-J in the final Figure.

Referee #2 (Comments on Novelty/Model System):

The mouse model is adequate to study influenza A virus infection. The authors use mainly PR8 virus, which is highly mouse adapted and used in many laboratories in mice.

Referee #2 (Remarks):

The authors show that OX40L-deficient mice are more resistant to challenge with PR8 virus compared to wild type mice. This correlates with reduced survival and cellular infiltration in the lungs. Bone marrow transplantation experiments provide some evidence that loss of OX40L expression on stromal cells is responsible for the increased resistance to influenza virus infection, although reduced cellular infiltration in OX40L^{-/-} mice is not observed in these experiments. M2-staining of lung sections is reduced in OX40L knockout mice. Influenza virus infection leads to an increase of the number of bronchiolar progenitor cells and a decrease in the number of club cells. OX40L mRNA levels are upregulated in BP cells following influenza virus infection. Similar findings are shown for an H3N2 virus infection. Infection of OX40L negative mice is associated with reduced levels of NP mRNA in ex vivo isolated cells and reduced M2 expression. BP and Club cells both bind SN lectin, suggesting the presence of alpha 2-6 linked sialic acid residues on their cell surface. Transfection of MDCK cells with an OX40L expression vector is associated with increased production of NP mRNA and M2 expression after influenza virus infection. Conversely, a mAb directed against OX40L reduces NP expression and partially protects mice against PR8 challenge. Site specific mutagenesis experiments and transfection of MDCK cells suggest that the N-glycosylation site at position 90 is important for OX40L-dependent increased influenza gene expression. Immuno-histological stains of lung section from an influenza and a bacterial pneumonia victim are shown in an attempt to document the clinical relevance of OX40L expression for the outcome of influenza virus infection in human.

The main message of the paper is that OX40 ligand, which is type II membrane protein that has important T cell co-stimulatory functions, serves as a receptor on bronchial epithelial progenitor cells for influenza A virus infection. Loss of OX40L, blocking OX40L with mAb or removing certain N-glycans on OX40L would then lead to reduced influenza A virus burden and associated disease. This is novel and interesting concept. However, the authors seem to ignore the role of OX40L in immune cell stimulation as a possible explanation for the increased survival in the knock out mice. In addition, their findings are not in line with published data on the lack of a phenotype for influenza in mice that lack alpha 2,6 sialyltransferase. Finally, at no point is live virus replication determined. I recommend that the authors perform extra experiments to strengthen their case.

Major remarks

1. The authors rely on RT-PCR for NP and on immunostaining of M2 to document "viral burden". The qRT-PCR data should be improved because they rely on a single house keeping gene for normalization. I recommend the authors to seek advice for improving the qRT-PCR data in Derveaux et al, Methods, 2010. In addition, the authors should determine newly produced virus from infected mice (wt and OX40L^{-/-} and MDCK cells transfected with OX40L or empty vector) by

plaque assay or by quantifying cytopathic effects in susceptible cells. MDCK cells are typically used for this.

2. Ox40L is a co-stimulatory molecule for T cells. An explanation for the increased survival of Ox40L^{-/-} mice could be a reduced level of pro-inflammatory cytokines. In fact, the reduced cellular infiltration in the BAL (Fig 1B) suggests that this may be implicated. The authors should compare the amounts of a number of cytokines such as IL4, IL6 and IFN gamma in BAL after infection in wt and knock out mice.

3. The findings are at odds with the paper by Glaser et al (Virus Res. 2007) that shows no significant impact of loss of alpha 2,6 sialic acid on N-glycans on susceptibility of mice to human influenza A virus infection. This should be discussed by the authors.

4. Why is there no more difference in cell infiltration in the radiation chimeras (Fig 1E)?

5. Panels in Fig 4D and 5 reveal very little. Please improve.

6. The authors conclude that club cells are poorly infected by influenza (Fig 4A-E). However, it was recently reported that these cells do become infected with influenza virus (Heaton et al JEM, 2014). Please refer to this paper and comment on it.

7. Fig 7: MDCK cells are highly susceptible to influenza virus infection. Why is it that the pNull control transfected cells do not produce virus? This is unexpected and may be due to the induction of type I IFN by the transfection procedure. Please determine IFN levels. Reduced expression of e.g. the Asn90 mutant could equally explain the reduced infectibility of the cells rather than the lack of a N-glycosylation at position 90 (Fig 7G).

8. The immunostains of the clinical samples in Fig 7 have a very poor resolution. The Ox40L staining should be performed with a negative control antibody and ideally an influenza A virus antigen (NP should work) antibody. Micrographs should be prepared with a much better resolution.

9. Please explicitly indicate whether littermates were used or not in the comparison of wt and ko mice. This is very important because the (gut) microbiota and microbiome composition is known to affect susceptibility to influenza virus infection.

Minor remarks:

It is hard to follow the figure numbering in the text. E.g. Fig 3D appears at the end of the results section. Please use ascending figure numbering in the text.

Resubmission

16 December 2015

Referee 1

Referee: (Comments on Novelty/Model System) “Some additional experiments, and repeating some experiments, as specified in the remarks to the author, should enhance the conclusions in the paper and make it more convincing.”

Response: We appreciate the Referee’s comments. According to the specified remarks, we have performed additional experiments.

Referee: (Remarks) “This is a very interesting paper, suggesting that OX40L can be a receptor for influenza virus and aid infection of bronchial epithelial cells. In general the experiments are quite convincing but additional experiments are needed to solidify and enhance the conclusions.”

Response: We appreciate the Referee's comments.

Referee: "1. In Figure 1a, the authors suggest that OX40L^{-/-} mice are protected from death with high dose intratracheal PR8/H1N1 infection whereas OX40^{-/-} mice are not. This is perhaps the most important data as it leads the rest of the paper in supporting the notion that the any results on lung pathology are not simply a repeat of what has been published before, which claimed that OX40-OX40L interactions drive an exacerbated T cell response to flu virus that causes pathology (Humphreys et al, JEM, 2003). Although the data as presented look convincing, this experiment was apparently only 1 experiment with 12 mice per group, and it is not indicated how survival was measured (presumably a loss of a certain amount of body weight given animal welfare rules). There are some differences in BAL counts and histology that somewhat correlate with the survival curves, but these are not so different to be convincing that this would fully explain the survival results. In Fig. 1d body weight loss in BM chimeras is shown, with very little change between groups, again questioning how real the data are in Fig. 1a. Thus, the contention that OX40^{-/-} mice differ from OX40L^{-/-} mice needs to be substantiated, with more repeat experiments, showing weight loss curves as well as survival curves. Importantly this should be performed with different doses of the virus to show the result does not simply apply to one specific situation."

Response: We agree with this comment. We have repeated the experiments with different doses of the virus. These data are shown in the revised manuscript (page 5, paragraph 1, lines 5-7, Figure S3).

Referee: "2. The authors suggest that OX40L is induced on bronchiolar progenitors by both H1N1 infection (Fig. 2H) and H3N2 infection (Fig. 3B) but the latter results are not convincing that OX40L is significantly or strongly upregulated. Again, as the conclusion of the paper hinges on OX40L being active on these cells independent of OX40, experiments should also be performed assessing weight loss and survival in OX40L^{-/-} and OX40^{-/-} with H3N2 to show that the results are not simply specific for one strain of flu."

Response: As suggested, we have included the relevant data of H3N2 in the revised manuscript (page 5, paragraph 1, lines 5-7, Figure S2).

Referee: "3. The authors imply that OX40L^{-/-} mice are protected from lethal flu infection in part because bronchiolar progenitor cells cannot be infected equivalently due to OX40L directly binding the virus. They show M2 expression in WT vs OX40L^{-/-} mice (Fig. 4G). If this is significant, and again not related to an immune response involving OX40-OX40L interactions, a kinetic analysis of levels of total virus particles (conventional plaque assay) as well as M2 protein in the lungs of OX40L^{-/-} and OX40^{-/-} mice over the first week of infection should also be dramatically different. Performing the experiment with H1N1 and H3N2 would again substantiate the conclusions."

Response: We agree with the Referee's comment. We have added data of conventional plaque assay in the revised manuscript (page 9, paragraph 1, lines 8-10, Figure S6).

Referee: "4. There is a disconnect in the relative NP expression in mouse OX40L transfected cells in Fig. 6A vs 6D. Can the authors explain why the qPCR results are substantially different from the semi-qPCR."

Response: The Referee's question is valid. In the original manuscript, the expression levels of the influenza virus NP gene were shown as relative to pNull-transfected cells of Fig. 6A (Fig. 4C in the revised manuscript) and control antibody-pretreated cells of Fig. 6D (Fig. 4F in the revised manuscript). This confusing notation could have raised the Referee's question, because OX40L-transfected cells of Fig. 6A and control antibody-pretreated cells of Fig. 6D should have the similar levels. We have redrawn Fig. 6A and shown the expression level as relative to OX40L-transfected cells (revised manuscript, Figure 4C).

Referee: "5. In Fig. 7, the authors present data with human OX40L transfected cells treated with sialidase. It would also be useful to perform the same study with mouse OX40L."

Response: We agree with the Referee's point. We have performed the additional experiment of mouse OX40L-transfected cells treated with sialidase (revised manuscript, page 10, paragraph 2, lines 13-14, Figure S7).

Referee: "6. The authors need to control for OX40L transfection in MDCK cells. Based on data in Fig. 8, it is not clear how uniform OX40L is expressed and the implication is that only a subset of cells may have expressed high amounts. This could impact the viral infection data. Also the data with the mutant OX40L constructs can only be evaluated effectively if similar levels of OX40L are expressed on the cell surface with each mutant. What are the absolute levels of OX40L on transfected cells and how do they compare to bronchiolar progenitors."

Response: We agree with the Referee. In the revised manuscript, we have added data to compare the expression levels of human OX40L on MDCK cells that were transfected with mutant human OX40L genes as well as the wild-type one (revised manuscript, page 11, paragraph 3, lines 5-6, Figure S10).

Referee: "7. The final data staining OX40L in tissue sections from autopsy are not convincing. The data are not clear and it is impossible to see any real staining. Moreover, bronchial epithelial cells are notoriously known to stain with almost any antibody. There is no control for the antibody to show it stains anything specifically. The comparison between tissues is also not valid, and making conclusions from tissues from single patients whose histories are not documented also calls into question any conclusions drawn."

Response: We agree. Since the data staining OX40L in tissue sections from autopsy are not convincing to draw any conclusions, we have deleted them in the revised manuscript.

Referee: (Minor point) "Some of the data are not logically placed. e.g. the first data on sialic acids (Fig. 4H and I) would be better in the current Fig. 6. There is no need for a separate Fig. 5 as this data is related to that in Fig. 4. Similarly, data in Fig. 3C and D are not mentioned until Fig. 7 is discussed and would be better in Fig. 7. Also, data in Fig. 8 should either be in a supplementary Figure, or data in Fig. 8 combined with Fig. 7G-J in the final Figure."

Response: We agree with the Referee. We have assembled Fig. 4H, 4I, and Fig. 6 of the original manuscript into new Fig. 4 of the revised manuscript. Also we have combined Fig. 5 of the original manuscript into Fig. 3 of the revised manuscript. To place logically Fig. 3 and Fig. 8 of the original manuscript, we have moved them to Fig. S4 and Fig. S11, respectively, in the revised manuscript.

Referee 2

Referee: (Comments on Novelty/Model System) "The mouse model is adequate to study influenza A virus infection. The authors use mainly PR8 virus, which is highly mouse adapted and used in many laboratories in mice."

Response: We appreciate the Referee's comments.

Referee: (Remarks) "The authors show that OX40L-deficient mice are more resistant to challenge with PR8 virus compared to wild type mice. This correlates with reduced survival and cellular infiltration in the lungs. Bone marrow transplantation experiments provide some evidence that loss of OX40L expression on stromal cells is responsible for the increased resistance to influenza virus infection, although reduced cellular infiltration in OX40L^{-/-} mice is not observed in these experiments. M2-staining of lung sections is reduced in OX40L knockout mice. Influenza virus infection leads to an increase of the number of bronchiolar progenitor cells and a decrease in the number of club cells. Ox40L mRNA levels are upregulated in BP cells following influenza virus infection. Similar findings are shown for an H3N2 virus infection. Infection of Ox40L negative mice is associated with reduced levels of NP mRNA in ex vivo isolated cells and reduced M2 expression. BP and Club cells both bind SN lectin, suggesting the presence of alpha 2-6 linked sialic acid residues on their cell surface. Transfection of MDCK cells with an Ox40L expression vector is associated with increased production of NP mRNA and M2 expression after influenza virus infection. Conversely, a mAb directed against Ox40L reduces NP expression and partially protects

mice against PR8 challenge. Site specific mutagenesis experiments and transfection of MDCK cells suggest that the N-glycosylation site at position 90 is important for Ox40L-dependent increased influenza gene expression. Immuno-histological stains of lung section from an influenza and a bacterial pneumonia victim are shown in an attempt to document the clinical relevance of Ox40L expression for the outcome of influenza virus infection in human."

Response: We appreciate the Referee's comments.

Referee: "The main message of the paper is that Ox40 ligand, which is type II membrane protein that has important T cell co-stimulatory functions, serves as a receptor on bronchial epithelial progenitor cells for influenza A virus infection. Loss of Ox40L, blocking Ox40L with mAb or removing certain N-glycans on Ox40L would then lead to reduced influenza A virus burden and associated disease. This is novel and interesting concept. However, the authors seem to ignore the role of Ox40L in immune cell stimulation as a possible explanation for the increased survival in the knock out mice. In addition, their findings are not in line with published data on the lack of a phenotype for influenza in mice that lack alpha 2,6 sialyltransferase. Finally, at no point is live virus replication determined. I recommend that the authors perform extra experiments to strengthen their case."

Response: We appreciate the Referee's comments.

Referee: (Major remarks) "1. The authors rely on RT-PCR for NP and on immunostaining of M2 to document "viral burden". The qRT-PCR data should be improved because they rely on a single house keeping gene for normalization. I recommend the authors to seek advice for improving the qRT-PCR data in Derveaux et al, Methods, 2010."

Response: We agree with this comment. We have performed the additional experiment using normalization against 2 more reference genes, and have achieved similar results (revised manuscript, page 7, paragraph 2, lines 9-11, Figure S5).

Referee: "In addition, the authors should determine newly produced virus from infected mice (wt and Ox40L-/- and MDCK cells transfected with Ox40L or empty vector) by plaque assay or by quantifying cytopathic effects in susceptible cells. MDCK cells are typically used for this."

Response: We agree and have performed the experiments of plaque assay. The data are shown in the revised manuscript (page 9, paragraph 1, lines 8-10, Figure S6; page 11, paragraph 2, lines 7-11, Figure S8).

Referee: "2. Ox40L is a co-stimulatory molecule for T cells. An explanation for the increased survival of Ox40L-/- mice could be a reduced level of pro-inflammatory cytokines. In fact, the reduced cellular infiltration in the BAL (Fig 1B) suggests that this may be implicated. The authors should compare the amounts of a number of cytokines such as IL4, IL6 and IFN gamma in BAL after infection in wt and knock out mice."

Response: We agree with the Referee. We have evaluated the amounts of cytokines suggested by the Referee. These data are shown in the revised manuscript (page 4, paragraph 2, line 9-page 5, paragraph 1, line 5, Figure S1).

Referee: "3. The findings are at odds with the paper by Glaser et al (Virus Res. 2007) that shows no significant impact of loss of alpha 2,6 sialic acid on N-glycans on susceptibility of mice to human influenza A virus infection. This should be discussed by the authors."

Response: As suggested, we discussed this issue in the revised manuscript (page 16, paragraph 1, lines 5-8).

Referee: "4. Why is there no more difference in cell infiltration in the radiation chimeras (Fig 1E)?"

Response: This is probably due to irradiation of the recipients and reconstruction of the immune system in engraftment of bone marrow. This point has been discussed in the revised manuscript (page 14, paragraph 1, lines 11-14).

Referee: “5. Panels in Fig 4D and 5 reveal very little. Please improve.”

Response: We agree with the Referee, and have improved them in Figure 3D and 3F of the revised manuscript.

Referee: “6. The authors conclude that club cells are poorly infected by influenza (Fig 4A-E). However, it was recently reported that these cells do become infected with influenza virus (Heaton et al JEM, 2014). Please refer to this paper and comment on it.”

Response: We agree with the Referee’s comment. We have referred the paper and commented on it in the revised manuscript (page 15, paragraph 1, lines 1-5).

Referee: “7. Fig 7: MDCK cells are highly susceptible to influenza virus infection. Why is it that the pNull control transfected cells do not produce virus? This is unexpected and may be due to the induction of type I IFN by the transfection procedure. Please determine IFN levels. Reduced expression of e.g. the Asn90 mutant could equally explain the reduced infectibility of the cells rather than the lack of a N-glycosylation at position 90 (Fig 7G).”

Response: We agree with the Referee. We have examined the expression levels of IFN- α and IFN- β in naive MDCK cells as well as human OX40L-, 90Asn→Ala mutant-, and pNull-transfected MDCK cells, and found no significant differences between naive and transfected cells. The data are shown in the revised manuscript (page 11, paragraph 2, lines 11-13, Figure S9). Moreover, to address the concerns about interference of the transfection procedure, we have specified that the transfected MDCK cells were washed twice before the influenza infection (revised manuscript, page 19, paragraph 1, lines 5-6).

Referee: “8. The immunostains of the clinical samples in Fig 7 have a very poor resolution. The Ox40L staining should be performed with a negative control antibody and ideally an influenza A virus antigen (NP should work) antibody. Micrographs should be prepared with a much better resolution.”

Response: According to the relevant comment of Referee 1, we have deleted the immunostaining of the clinical samples in the revised manuscript.

Referee: “9. Please explicitly indicate whether littermates were used or not in the comparison of wt and ko mice. This is very important because the (gut) microbiota and microbiome composition is known to affect susceptibility to influenza virus infection.”

Response: As suggested, we have explicitly indicated that wild-type mice used in this study were not littermates of gene-deficient mice (revised manuscript, page 16, paragraph 3, lines 2-3).

Referee: (Minor remarks) “It is hard to follow the figure numbering in the text. E.g. Fig 3D appears at the end of the results section. Please use ascending figure numbering in the text.”

Response: We agree with the Referee. We have moved Fig. 3 of the original manuscript to Fig. S4 in supplementary data of the revised manuscript to keep the order of figures.

Thank you for the submission of your revised manuscript to EMBO Molecular Medicine. We have now received the enclosed reports from the referees that were asked to re-assess it. As you will see the reviewers are now globally supportive and I am pleased to inform you that we will be able to accept your manuscript pending editorial final amendments.

Please submit your revised manuscript within two weeks.

I look forward to reading a new revised version of your manuscript as soon as possible.

***** Reviewer's comments *****

Referee #1 (Remarks):

The authors have addressed the majority of the prior critiques satisfactorily.

Referee #2 (Comments on Novelty/Model System):

Compared to the original submission, this revised manuscript has improved considerably in technical quality. More controls are now included, infections were with different influenza A virus strains and doses and the quality of the micrographs has been improved. The indication that OX40L is a putative receptor for influenza A viruses is well substantiated.

Referee #2 (Remarks):

The experiments performed for the revision are well performed and strengthen the paper. It is a pity that immunostaining of the clinical sample could not be improved and had to be withdrawn.

Corresponding Author Name: Toshiaki Kikuchi
 Journal Submitted to: EMBO Molecular Medicine
 Manuscript Number: EMM-2015-05037